# Evolution Analysis of Free Fatty Acids and Aroma-Active Compounds during Tallow Oxidation

**DOI:** 10.3390/molecules27020352

**Published:** 2022-01-06

**Authors:** Shiqing Song, Feiting Zheng, Xiaoyan Tian, Tao Feng, Lingyun Yao, Min Sun, Lei Shi

**Affiliations:** 1School of Perfume and Aroma Technology, Shanghai Institute of Technology, Shanghai 201418, China; sshqingg@163.com (S.S.); zhengft0204@163.com (F.Z.); a819036058@126.com (X.T.); ft422@sina.com (T.F.); Lyyao@sit.edu.cn (L.Y.); 2Pudong New Area Agro-Technology Extension Center, Shanghai 201201, China; shileilyac@eyou.com

**Keywords:** oxidized tallow, aroma-active compounds, solvent-assisted flavor evaporation, gas chromatography-olfactometry, partial least-squares regression

## Abstract

To explore the role of fatty acids as flavor precursors in the flavor of oxidized tallow, the volatile flavor compounds and free fatty acid (FFAs) in the four oxidization stages of tallow were analyzed via gas chromatography (GC)–mass spectrometry (MS), the aroma characteristics of them were analyzed by GC–olfactory (GC-O) method combined with sensory analysis and partial least-squares regression (PLSR) analysis. 12 common FFAs and 35 key aroma-active compounds were obtained. Combined with the results of odor activity value (OAV) and FD factor, benzaldehyde was found to be an important component in unoxidized tallow. (*E*,*E*)-2,4-Heptadienal, (*E*,*E*)-2,4-decadienal, (*E*)-2-nonenal, octanal, hexanoic acid, hexanal and (*E*)-2-heptenal were the key compounds involved in the tallow flavor oxidation. The changes in FFAs and volatile flavor compounds during oxidation and the metabolic evolution of key aroma-active compounds are systematically summarized in this study. The paper also provides considerable guidance in oxidation control and meat flavor product development.

## 1. Introduction

Lipid oxidation or decomposition can easily lead to an undesirable flavor. Nevertheless, the oxidative degradation of lipid is essential for the development of the characteristic meat flavor [1,2]. Many of the compounds in meat are lipid oxidation and free radical reaction products all of which play an important role in the formation of the distinctive flavor characteristics of meats such as beef, pork, and lamb [3]. In other words, leanness is responsible for the basic meaty flavor common to all meat types, whereas volatile compounds sourced from different types of lipids can provide meat its characteristic flavor [4] (pp. 210–230).

The mechanism underlying characteristic meat flavor precursor formation via lipid oxidation has been the focus of recent food flavor research. Many researchers have explored the importance of lipid oxidation and degradation in characteristic flavor formation in meat products under different processing conditions. These meat products have included Beijing duck, marinated pork meat, and Chinese bacon [5,6,7]. Oxidative lipid degradation produces many volatile compounds, and hundreds of volatile compounds have been identified in heated beef fat. In general, these compounds are produced via fatty acid oxidation [8,9]. Andrés found that the fatty acids produced from lipolysis might be the main flavor precursors here [10]. Moreover, Varlet found that in fish, fatty acid and triglyceride oxidization led to the production of hydroperoxides, which then degraded to produce many saturated or unsaturated aliphatic aldehydes the most important hydroperoxide decomposition products [11]. Aldehydes were also found to be the most significant flavor compounds produced via lipid decomposition [12]. For instance, nonanal is the most significant aldehyde produced via oleic acid oxidation [13]. 1-Octen-3-ol, which has a strong mushroom flavor, can be derived from the oxidative breakdown of linoleic acid [14].

Unoxidized lipid does not produce an obvious characteristic meat flavor in the thermal reaction, whereas overoxidized lipid affords an unpleasant odor such as that of rancidity. Therefore, only moderately oxidized lipid is conducive to the overall presentation of the characteristic meat flavor [15,16]. Lipid oxidation rate and degree depend on fatty acid composition and concentration. As free fatty acid concentration increases, lipids are further oxidized to produce numerous different volatile compounds [7,17,18]. The oxidation rate of free fatty acids has been found to be higher than that of bound fatty acids in glycerides [12]. The more the free fatty acids (FFAs) in the lipid, the faster is its spontaneous oxidation rate. However, FFA concentrations in tallow are usually very low, and thus, they need to be predissociated such that the oxidation reaction occurs efficiently. Therefore, we previously adopted enzymatic hydrolysis-mild thermal oxidation: Exogenous lipase was used for controlled enzymolysis of tallow in advance, followed by controlled oxidation under mild heating conditions (<120 °C) [19]. Compared with oxidation with heating at high temperatures, the aforementioned method demonstrated higher controllability, with the oxidation products being more abundant. In addition to enzymolysis–oxidation products, lipase hydrolysates and direct oxidation products were present in the system. However, previous studies have only demonstrated that lipase hydrolysis–mild oxidation products affect the enhancement of meat flavor aroma [19]. Further analysis on how different oxidation levels affect the change of flavor precursors and whether there is quantitative effect relationship between characteristic aroma components and precursor compounds is required.

In this study, solid phase extraction (SPE) combined with gas chromatography (GC)–mass spectrometry (MS) was used to conduct qualitative and quantitative analysis of FFAs in the oxidation process of tallow. Moreover, solvent-assisted flavor evaporation (SAFE) combined with GC-MS and with GC–olfactometry (GC-O) were used to determine the key aroma-active compounds in the oxidized tallow. Partial least-squares regression (PLSR) analysis was performed on the basis of the abovementioned experimental results combined with sensory evaluation scores for studying the formation mechanism of characteristic flavor of oxidized tallow, exploring the quantitative effect between characteristic meat flavor precursor compounds, and elucidating the formation pathways of the different flavor precursors and the underlying mechanisms. The changes in FFAs and volatile compounds in the four oxidation process of tallow were analyzed, the key aroma-active compounds were screened, and their metabolic evolution was summarized.

## 2. Results and Discussion

### 2.1. Changes in FFAs during Oxidation

FFA oxidation rate is higher than that of bound fatty acids in glycerides [12]. The more the FFAs in the lipid, the faster is the spontaneous oxidation rate. Therefore, here, enzymatic hydrolysis of tallow was performed in advance so as to increase the efficiency of the subsequent oxidation reaction. FFA contents and compositions at different oxidation stages of refined tallow treated via enzymatic hydrolysis under the same conditions are listed in Table 1. In total, 12 common FFAs and 3 trans fatty acids were detected. C16:0, C18:0 and C18:1 were the primary FFAs in the tallow samples with concentrations >1.46 mg/g both before and after oxidation and thus more abundance compared with other FFAs.

Table 1 and Appendix A indicate that the total amount of FFAs is inversely proportional to the oxidation time of tallow, saturated fatty acids (SFAs) are the main component, followed by monounsaturated fatty acids (MUFAs). Compared with T0, the total amount of FFAs decreased by 19.19%, 37.42%, and 67.68% in T1, T3, and T5 respectively. Total SFAs, MUFAs, and polyunsaturated fatty acids (PUFAs) demonstrated a tendency similar to that of total FFAs during all the stages. Among them, the concentration of MUFAs and PUFAs continued to decrease due to oxidative degradation and formation of volatile aroma compounds. Also, the rupture of carbon–carbon bond of saturated fatty acids also occurs at high temperature. Short-chain fatty acids come from the oxidative degradation of long-chain SFAs or polyunsaturated fatty acids (UFAs) or the secondary degradation of primary degradation products, and the concentration change of volatile short-chain fatty acids (such as propanoic acid, butanoic acid and pentanoic acid) can confirm this conjecture. Moreover, UFAs may be oxidized to increase the SFAs content during lipid oxidation, which is consistent with the results data [20].

The decreases in FFA contents were attributable to FFA oxidation a tendency in line with the law of oxidation. Compared with T0, T1 demonstrated a significant decrease in the contents of C14:0, C16:0, C16:1 n-7, C17:0, C18:0, C18:1 n-9 and C18:2 n-6, especially unsaturated fatty acid contents. In this oxidation process, FFAs are more likely involved in oxidation, producing large amounts of hydroperoxides. Hydroperoxides are identified as the most conspicuous early lipid oxidation products. However, these compounds exist for a very short time, and their breakdown is as fast as or even faster than their formation [21]. After 1 h, the changes in the tallow oxidation reaction (between T0 and T1) are significantly different. The contents of FFAs such as C10:0, C11:0, C16:0, C18:0, C18:1 n-9 and C18:2 n-6 decreased significantly. Similarly, many volatile components including seven alcohols, seven aldehydes, six ketones, four acids and four esters were produced. When the oxidation time extended, hexanal, heptanal, (*E*)-2-hexenal, octenal, (*E*)-2-nonenal, and hexanoic acid contents increased continuously. In particular, octenal contents were 44.04 times higher at 3 h than at 1 h. Therefore, these volatile components are likely produced via the oxidation of these FFAs. Of these components, hexanal is often regarded as a sign of oxidation and is generally characterized by fatty and green descriptors [21]. It can appropriately reflect the extent of oxidation in products rich in ω-6 fatty acids [22]. (pp. 519–542). After 3 h of oxidation, FFA oxidation degree, especially that of C16:0, C17:0, C18:0, and C18:1 n-9, increased significantly. During oxidation from T3 to T5, C17:0, C18:0, and C18:1 n-9 contents decreased by 76.7%, 66.4%, and 59.1%, respectively. No significant change was noted in the types of volatile components.

### 2.2. Sensory Analysis

Eight sensory attributes were selected for sensory evaluation of oxidized tallow: milky, cheesy, sweet, beef tallow, fatty, green, oxidized, and smell of mutton (Figure 1). The results of sensory evaluation reflect the flavor changes during the tallow oxidation process. ANOVA of the interactions showed nonsignificant differences in the sample replicates and between the sensory panels, indicating that the sensory data were valid (Table 2). However, a significant interaction between sample and panelist was observed for the milky, sweet, fatty, green, oxidized, and mutton smell attributes (all *p* ≤ 0.001) indicating that the panelists probably used different criteria to express their observations that reflected their personal or physiological differences in perceived intensity of scoring.

Due to the aforementioned significant interaction, an adjusted F-test was performed. The panelists’ observations were treated as random effects, and the adjusted F-test was performed using the mean square of sample × panelist interaction instead of mean square of error as the denominator [12]. As shown in Table 2, the ANOVA results after taking the variation of panelists into account showed a significant difference between four samples for all attributes (all *p* ≤ 0.001), and the aroma types showed different dominant sensory properties at different oxidation stages.

The intensities of the milky, cheesy, fatty, green, and oxidized attributes were significantly higher in T3 and T5 samples than in T0 and T1 samples (Figure 1). In particular, milky, cheesy, and fatty aroma had the highest scores in T3 samples which then decreased gradually with the prolonging of oxidation time. The changes should be related to changes in compounds that exhibit fatty aroma characteristics during oxidation, of which aldehydes and acids were the main contributors. However, the intensity of green and oxidized attributes were proportional to the oxidation time. In contrast, the smell of mutton scores decreased with the prolongation of the oxidation process, which may be related to fatty acid contents in tallow. Sanudo, Enser and Campo indicated that the sources of the smell of mutton were C18:0 and C18:3 [23].

### 2.3. Analysis of Aroma-Active Compounds of Oxidized Tallow

Fatty food aroma and meat flavor originate from the volatile compounds directly produced via lipid oxidation, and the other part is obtained with the participation of lipid oxidation products in Maillard reaction. Therefore, the study of volatile flavor compounds in tallow with different oxidation levels is a necessary precondition to explore the sources of characteristic aroma in beef.

#### 2.3.1. GC-MS Analysis

SAFE combined with GC-MS was used to analyze flavor compounds in four oxidation stages of tallow. The distribution of volatile flavor compounds were found to vary significantly in the different oxidation stages (Figure 2a–c). In total, 46 volatile flavor compounds were detected. Of the four samples, 23, 14, 42 and 35 volatile flavor compounds were quantified, respectively (Figure 2a). They included 7 alcohols, 13 aldehydes, 10 ketones, 6 acids, 4 lactones, 1 volatile phenol, 1 furan, and 2 terpenes. As shown in Figure 2b, the volatile flavor compounds in four oxidation stages showed significant differences. At early oxidation stages (T0 and T1), the volatile compound concentrations decreased slightly. Compared with T0, the volatile flavor compounds in T1 decreased by 43.4%. The primary stage of fatty acid oxidation is the formation of hydroperoxide, which is unstable and and easily decomposed [21]. Therefore, the formation of hydroperoxide more than decomposition may be one of the reasons for the decrease of compound content in this stage. Existing volatile compounds may also undergo polymerization, interaction and loss during oxidation, which may lead to reduction. As oxidation progressed, these concentrations demonstrated a sharp increase after 1 h of oxidation. However, 3 h later, the concentration showed a downward trend with a decrease of 26.5%. After heating for another 2 h, the concentrations of most volatile compounds increased significantly. These results are consistent with those of our previous study: 2–3 h was more suitable for the controlled oxidation of tallow [15]. Aldehydes and acids were the main volatile aroma-active compounds of oxidized beef tallow during the oxidation process. They accounted for 15.8–69.2% and 8.35–57.5% of the total volatile compounds, respectively. Compared with that in T0 samples, the number of aldehydes increased 15.1 times in T3 samples. Acids, furans, and ketones also reached their maximum concentrations in T3 samples, consistent with the changes in volatile aroma-active compounds. Alcohols and esters were detected in the T3 and T5 samples but not in the T0 and T1 samples.

To present the material differences of tallow in different oxidation stages, heatmap analysis was employed in Figure 2c. Here, among the four oxidation stages, T0 and T1 were clustered into one class, whereas T3 and T5 were clustered into another. In the red (high concentration) region of the T0 samples, five compounds including one aldehyde, two acids, one terpene, and one phenol were clustered. Most of these compounds did not occur in other samples, indicating their critical roles in the T0 sample. Moreover, 38 compounds were clustered in the red (high concentration) region of T3 sample, most of which did not occur in the T0 and T1 samples. Moreover, they represented significant volatile flavor compounds produced after tallow oxidation for 1 h. In general, T5 samples had a volatile flavor compound composition similar to those of T3 samples, but most of them had slightly lower concentrations.

#### 2.3.2. GC-O Analysis

In the four oxidation stages, 35 aroma-active compounds with an FD factor of ≥1 were detected via GC-O analysis (Table 3). Of these, 15 were detected in T0, including nine aldehydes, five acids and one furan. Among them, 11 had FD factor > 4. Benzaldehyde (64) had the highest FD factor and a sweet, fruity aroma. In sensory evaluation, T0 had the highest score in sweet attribute. Thus, benzaldehyde may be an important source of sweet attribute in T0. n-Decanoic acid has the higher FD factor in T0, and it was difficult to detect in diluted samples in T1, T3, and T5. These volatile compounds, which contributed sweet and fatty aroma, were generally the main compounds responsible for the typical aroma in unoxidized tallow (T0).

Compared with T0, T1 demonstrated many compounds with increasing FD factors including (*E*)-2-nonenal (512), (*E*)-2-heptenal (64), octanal (64), hexanal (32), (*E*,*E*)-2,4-decadienal (32), hexanoic acid (32), dodecanoic acid (32) and octanoic acid (16). Their contribution to aroma increased to different extent. Notably, compared with T0, volatile flavor compound types and concentrations in the system decreased at this stage, but they played a more significant role, considering the variation of the FD factors in most aldehydes and acids. Hexanal (8→32), (*E*)-2-heptenal (8→64), (*E*,*E*)-2,4-heptadienal (0→256), (*E*,*E*)-2,4-decadienal (4→32), hexanoic acid (4→32), heptanoic acid (1→4), octanoic acid (8→16) and dodecanoic acid (2→32) present the same phenomenon. In this stage, hydroperoxide formation rate was higher than its decomposition rate, which may have led to a downward trend in compound species and contents. Although the species and contents of the compounds decreased, the FD factors of some compounds increased. Therefore, undetectable compounds (such as hydroperoxide) in the system may enhance the aroma of volatile flavor compounds.

Compared with T1, T3 showed significantly increased aldehyde concentrations, with aldehydes remaining the most predominant compound. The FD factors of (*E*,*E*)-2,4-heptadienal and (*E*,*E*)-2,4-decadienal increased to 1024 and 512, respectively. (*E*)-2-Nonenal (512) which had a very significantly high FD factor in T1, did not change significantly in T3. In addition, many characteristic volatile flavor compounds were produced from T1 to T3. Of them, pentanoic acid (8), 2-pentylfuran (16), 1-pentanol (32), 2-octanone (32), 2-nonanone (32), 1-octen-3-ol (64), propanoic acid (64), 1-octanol (64) and butanoic acid (64) appeared with high FD factors.

From T3 to T5, the FD factors of most volatile flavor compounds began to decrease consistent with the changing trend of their concentrations. At the same time, the FD factors of some volatile flavor compounds, such as (*E*)-2-octenal (8→32), 1-heptanol (4→256), γ-valerolactone (0→4), γ-hexalactone (2→4), γ-octalactone (0→4), dodecanoic acid (0→4) and tetradecanoic acid (64→128), increased gradually.

### 2.4. OAV Analysis

In total, 35 aroma-active compounds (with FD factors of ≥1) were selected and quantitated (Table 4). OAVs of these aroma-active compounds were calculated to investigate their contributions to the overall aroma of the oxidized tallow.

In T0, (*E*,*E*)-2,4-decadienal showed the highest OAV (=421,063). Moreover, the OAV of octanal, heptanal, (*E*)-2-nonenal, hexanal, 2-pentylfuran, nonanal, (*E*)-2-hexenal, (*E*)-2-heptenal and dodecanoic acid was >1000. Benzaldehyde was only detected in T0. Aldehydes, except benzaldehyde, have extremely low thresholds, and the OAVs of these aroma-active compounds were >100. Among the fatty acids, the short chain fatty acids might contribute to the special flavour characteristic, because of their low odour threshold values [15]. But compared with aldehydes, acids generally have higher thresholds [24]. However, because our oxidized tallow had many acids at high concentration, the OAV of these acids was >1. In T0, hexanoic acid, heptanoic acid, octanoic acid, nonanoic acid and benzaldehyde played an important role, combining with the rich and various aroma-active compounds to finally form the characteristic oxidized tallow aroma in T0.

Except for D-limonene (OAV = 167) and (*E*)-2-nonenal (OAV = 56,751), the OAVs of many compounds in T1 were lower than their OAVs in T0. The concentration of aldehydes varied significantly (*p* < 0.05) from T0 to T1. The OAV of octanal decreased to 20,853, whereas that of heptanal also decreased to <20,000. The concentration of (*E*,*E*)-2,4-decadienal, which had the highest OAV in T0, decreased gradually with oxidation and could not be detected in T1. In T1, aldehydes such as (*E*)-2-nonenal, octanal and heptanal still played a major role, even if their concentrations were low.

Compared with T1, many aldehydes, alcohols, ketones and esters were detected in T3 at higher concentrations. In this sample, (*E*,*E*)-2,4-decadienal, (*E*)-2-nonanal, octanal, hexanal and 2-pentylfuran showed the highest OAVs which resulted in a mainly green and fatty aroma. In terms of sensory attributes, the T3 samples showed strong fatty, cheesy, milky and green attributes. (*E*,*E*)-2,4-Decadienal showed the highest OAV (=4,821,417). This was contradictory to the AEDA results: (*E*,*E*)-2,4-heptadienal (OAV = 1803) has the highest FD factor, possibly because OAV using the threshold of (*E*,*E*)-2,4-heptadienal was calculated in air. Moreover, this phenomenon revealed the complexity of the food system and potential influences of the food matrix. In addition, many compounds had high OAV: (*E*)-2-nonenal had an OAV of >1,000,000. 1-Octen-3-ol, 1-heptanol, hexanal, decanal and 2-pentylfuran had an OAV of 100,000–1,000,000. Hexanal, 2-pentylfuran and nonanal had an OAV of 10,000–100,000. (*E*)-2-Heptenal, γ-octalactone and (*E*)-2-hexenal had an OAV of 1000–10,000. Aldehydes remained the predominant compound even in T3 and combined with a rich variety of aromatic active compounds to form the characteristic oxidized tallow flavor.

After 3 h of oxidation, volatile flavor compound concentrations showed a slightly decreasing trend, consistent with the trend of changes in the FD factor. OAV analysis showed that (*E*,*E*)-2,4-decadienal contributed the most to the aroma during the oxidation process, followed by (*E*)-2-nonenal, octanal and heptanal. Notably, although heptanal had a high OAV, its FD factor remained at 4–8, and its contribution was lower than that of some compounds with relatively low OAVs, which revealed the potential impact of the food matrix [25].

### 2.5. Evolution Analysis of Aroma-Active Compound in Oxidized Tallow

By combining the results of GC-MS and GC-O as well as considering the changes in FD factors and OAVs, volatile flavor compound sources became traceable, and these were then analyzed.

Aldehydes, with significant influences on meat aroma by their low odor thresholds, have been reported to be closely related with lipid oxidation [6]. 12 aldehydes with high aroma activity were identified in the oxidation process of tallow. Most of them had fatty and green aroma, whereas few of them had sweet, herbal, fruity and oily aroma. Benzaldehyde was detected only in T0, and its sweet aroma was a significant source of the sweet attribute in the T0 samples. Benzaldehyde has been reported in unsmoked fish flesh and black pork, formed due to the degradation of α-linolenic acid to phenylacetaldehyde [11,26,27,28]. The AEDA results demonstrated that (*E*,*E*)-2,4-heptadienal, (*E*)-2-nonenal, (*E*,*E*)-2,4-decadienal, octenal, (*E*)-2-heptenal and (*E*)-2-heptenal were the key components of the characteristic flavor of tallow, especially in T3 and T5. Of the aforementioned compounds, (*E*)-2-nonenal, (*E*)-2-heptenal, (*E*,*E*)-2,4-heptadienal and (*E*,*E*)-2,4-decadienal are significant contributors of the beef flavor [15,19,29]. Studies have shown that long-chain MUFAs and PUFAs were the precursors of these aldehydes. (*E*,*E*)-2,4-Decadienal and (*E*)-2-nonenal are linoleic acid oxidation products, whereas (*E*,*E*)-2,4-heptadienal is a byproduct of the linolenic acid oxidative process [11,30]. Moreover, heptanal precursors have confirmed to be n-3 and n-6 PUFAs [30]. In addition, hexanal is oxidized from linoleic acid, whereas octanal and nonanal are derivatives of oleic acid oxidation [11].

The most pronounced increase was found for aliphatic aldehydes between 1 and 3 h, followed by that for aliphatic alcohols. 1-octen-3-ol imparts a strong mushroom aroma with lower odor threshold, could be a major contributor to the overall meat aroma [31,32]. In addition, 1-pentanol, 1-heptanol and 1-octanol have significant aroma activity. These key alcohols could not be detected at the initial stage of oxidation, indicating that they were the products of secondary oxidation, and most of the alcohols came from the oxidative degradation of FFAs.

Ketones are also important aroma-active compounds in oxidized tallow. They primarily result from amino acid degradation, lipid oxidation, and Maillard reaction [33]. Two ketones with higher aroma activity were detected, including 2-octanone (earthy) and 2-nonanone (fruity). These two ketones were derived from UFA oxidation, which showed the highest aroma activity in T3, played a coordinating and complementary role in the formation of the characteristic flavor of oxidized tallow [34].

Most acids have high thresholds and are difficult to perceive, the high concentrations in beef tallow sample make them sensible and important contributor of characteristic flavor. In total, 10 important acids were detected, their main contribution to aroma attributes were fatty and cheesy. These acids were mostly short-chain fatty acids, originating from the oxidative degradation of long-chain SFAs or UFAs or from the secondary degradation of the primary degradation products. Short-chain fatty acids (such as propionic acid, butanoic acid and pentanoic acid) were detected only in T3 and T5, so they are most likely generated via the oxidative degradation of long-chain fatty acids leading to the breaking of carbon–carbon bonds. Song et al. reported that the short-chain fatty acids with a low odor threshold value might contribute to the special flavor characteristic [15]. In addition, the fatty acids with a high odor threshold value (such as C8:0 and C9:0) might be the key precursors, which could affect meat flavor formation indirectly [35].

Lactones are lipid degradation products generated from the intramolecular dehydration of hydroxy fatty acids; some of these are positively correlated with sweet and fruity characteristics [36]. Many unsaturated lactones produce a pleasant frying aroma. Most of these lactones are produced via the thermal degradation of linoleic acid. γ-Heptalactone and γ-octanoiclactone are considered potent aromatic lipid degradation products in shallow-fried beef [37]. In addition, an important furan was detected. 2-Pentylfuran (fruity) has an extremely low threshold and is perceived easily. It has been reported that 2-pentylfuran was a potential indicator of heat stress or lipid peroxidation for duck meat before heat processing [18]. The compound isolated via GC-O had a slightly pungent fruity and green flavor, especially in T3. Its threshold in water was extremely low (=0.0058 mg/kg). In general, 2-pentylfuran produces a unique bean flavor at a concentration of 1–10 mg/kg derived from the autooxidation of C18:2 n-6 and catalyzed by C18:3 n-3.

### 2.6. Relationship of FFA and Aroma-Active Compound Data with Sensory Attribute Data

The correlation of FFAs and aroma-active compounds with sensory attributes in the four tallow oxidation stages were investigated by PLSR analysis. The relationship between 12 FFAs and 35 volatile flavor compounds in tallow samples and the sensory data of tallow sample is shown in Figure 3. FFAs were named as A1–A12 (Table 1), and volatile flavor compounds as 1–35 (Table 4). The X-matrix was designated as FFAs and volatile flavor compounds in tallow, and the Y-matrix as the oxidation tallow sample and sensory attribute data. PLSR provided a binary factor model explaining 93% of the X-variance (aroma-active compounds and FFAs) and 82% of the Y-variance (sensory attribute and oxidized tallow sample data). The variables marked with small circles were significant. All sensory attribute data, FFAs, most samples (except T1) and most volatile flavor compounds (except compound 35) were placed between the two ellipses.

As shown in Figure 3, T0 and T1 samples were on the left side of the plot, whereas T3 and T5 samples were on the right side. A result somewhat similar to the clustering results of the heat maps (Figure 2c). All sensory attributes were located between the two ellipses, which were significantly correlated with some FFAs and volatile flavor compounds. Thus, T0 samples correlated significantly with the beef tallow and smell of mutton attributes and positively with all identified FFAs but negatively with most of the volatile flavor compounds. This phenomenon might confirm that the beef tallow and smell of mutton attributes noted in lightly oxidized tallow (T0 and T1) were most likely caused by fatty acids corroborating the results a previous study [23]. It also confirmed that only enzyme or short-term oxidation treated tallow could not be acquired to develop desirable oxidized tallow characteristic flavor. The T1 samples and compound 35 were located in the inner ellipse, but without any significant correlations. This could be the reason that many compounds detected later were not formed in T1. The T3 samples were associated with milky, fatty and cheesy attributes corresponding to the sensory evaluation results in Figure 1 and Table 2, where T3 had the highest scores. These three attributes (milky, fatty and cheesy) were quite closely located, indicating that these sensory attributes covaried simultaneously with some GC-MS variables. These variables marked with small circles were determined to be significantly correlated with sensory attributes. There are 24 compounds on the right labeled with small circles that were predicted to be powerful aroma-active compounds that contribute to aroma properties. In the blue region corresponding to these three attributes, most compounds have the highest concentration in T3, followed by that in T5. This result indicated that the oxidation time has a significant effect on the oxidation products of tallow, and the products of oxidation for 3 h were better than those of oxidation for 5 h. Compared with the T3 samples, the T5 samples had a higher correlation with the oxidized and green attributes. This results indicated that prolonged oxidation time produces negative odors, such as oxidized.

On combining the GC-MS and GC-O results, several compounds with important roles in the generation of the oxidized tallow flavor were selected to speculate the quantitative relationship between them and their precursor compounds. Consequently, a model equation was established by PLS1 regression analysis:Y(27, (*E*,*E*)-2,4-Decadienal) = 2.432959 − 0.064279X_1_ − 0.062718X_2_ − 0.064358X_3_ − 0.054972X_4_ − 0.054372X_5_ − 0.052052X_6_ − 0.057320X_7_ − 0.047814X_8_ − 0.047904X_9_ − 0.053319X_10_ − 0.048686X_11_ − 0.054959X_12_(1)
Y(16, (*E*,*E*)-2,4-Heptadienal) = 3.233926 − 0.093577X_1_ − 0.091305X_2_ − 0.093692X_3_ − 0.080029X_4_ − 0.079155X_5_ − 0.075777X_6_ − 0.083446X_7_ − 0.069607X_8_ − 0.069739X_9_ − 0.077622X_10_ − 0.070877X_11_ − 0.080009X_12_(2)
Y(19, (*E*)-2-Nonenal) = 3.060672 − 0.086301X_1_ − 0.084205X_2_ − 0.086407X_3_ − 0.073806X_4_ − 0.073000X_5_ − 0.069885X_6_ − 0.076957X_7_ − 0.064195X_8_ − 0.064316X_9_ − 0.071586X_10_ − 0.065366X_11_ − 0.073787X_12_(3)
Y(9, Octanal) = 3.220554 − 0.088788X_1_ − 0.086632X_2_ − 0.088897X_3_ − 0.075933X_4_ − 0.075104X_5_ − 0.071899X_6_ − 0.079175X_7_ − 0.066045X_8_ − 0.066170X_9_ − 0.073650X_10_ − 0.067250X_11_ − 0.075914X_12_(4)
Y(18, Benzaldehyde) = −1.457575 + 0.077090X_1_ + 0.075218X_2_ + 0.077185X_3_ + 0.065928X_4_ + 0.065209X_5_ + 0.062426X_6_ + 0.068744X_7_ + 0.057343X_8_ + 0.057452X_9_ + 0.063946X_10_ + 0.058390X_11_ + 0.065912X_12_(5)

Here, Y is a key aroma-active compound in oxidized tallow samples, and X_1_–X_12_ denote the contents of the corresponding FFAs in Table 1. The results obtained with this partial least-squares regression based on the content of the 12 FFAs of tallow samples may predict the key aroma-active compound levels.

## 3. Materials and Methods

### 3.1. Materials

Refined tallow was obtained from Tianjin Yixing Halal Food Co. Ltd. (Tianjin, China). Lipase [Candida, (BR, 20,000 U/g)] was obtained from Shanghai yuanye Bio-Technology Co., Ltd. (Shanghai, China). Nonanal [reagent grade (RG), 95%], octanal (RG, 98%), acetic acid (RG, 99%), hexanoic acid (RG, 99%), heptanoic acid (RG, 98%+), dodecanoic acid (RG, 99%+), γ-valerolactone (RG, 99%+), γ-hexalactone (RG, 99%+), γ-heptalactone (RG, 98%+), nonanoic acid (RG, 98%+), D-limonene (RG, 99%), 1-pentanol (RG, 99%) and hexanal (RG, 98%) were purchased from Adamas Reagent Co., Ltd. (Shanghai, China). Benzaldehyde [≥99.5% (GC)], propanoic acid [≥99% (GC)], butanoic acid [analytical reagent (AR), 99%], pentanoic acid [>99.5% (GC)], octanoic acid [≥99.5% (GC)], Methyltridecanoate [≥97% (GC)], (*E*)-2-Heptenal [≥95% (GC)], 1-Octen-3-ol [≥98% (CP)], 2-octanone [≥98% (CP)], 1-octanol [≥95% (GC)], n-Decanoic acid [≥98% (CP)] were obtained from Aladdin (Shanghai, China). (*E*)-2-Hexenal [≥98% (GC)], tetradecanoic acid [≥99% (GC)] were obtained from Collins (Shanghai, China). (*E*)-2-Nonenal [≥95% (GC)] were purchased from Macklin (Shanghai, China). Tert-butyl methyl ether [≥99% (GC)] was purchased from Rhawn (Shanghai, China). γ-Octalactone (AR, 98%) was obtained from Acros (Shanghai, China). C7–C30 n-alkanes and 1,2-dichlorobenzene (GC) were obtained from Sigma-Aldrich Co., Ltd. (Shanghai, China). Dichloromethane (AR) and anhydrous sodium sulfate (AR) were purchased from Sinopharm Chemical Reagent Co., Ltd. (Shanghai, China). Heptanal [95% (GC)], (*E*)-2-Octenal [≥96% (GC)] was purchased from TCI (Shanghai, China). Hexane [≥97% (AR)], (*E*,*E*)-2,4-Decadienal [≥90% (AR)], (*E*,*E*)-2,4-heptadienal [≥90% (AR)], 1-penten-3-ol [≥97% (AR)], methyl alcohol [≥99% (AR)], isopropanol [≥99% (AR)], 2-nonanone [≥98% (AR)], 1-heptanol [≥99% (AR)], decanal [≥97% (AR)], NaH_2_PO_4_·2H_2_O [≥99% (AR)], Na_2_HPO_4_·12H_2_O [≥99% (AR)] purchased from Shanghai Titan Technology Co., Ltd. (Shanghai, China).

### 3.2. Samples Preparation

Enzymatic hydrolysis of tallow: Phosphate buffer solution (pH 7.5) was placed in a reaction flask in equal proportion to the refined tallow and stirred mechanically at 150 rpm. The enzyme was inactivated at 95 °C for 5 min, and 1.5% Candida lipase was added when the temperature in the reaction flask dropped to 45 °C. After 5 h of reaction, the sample was placed in a water bath at 95 °C for 5 min to inactivate the enzyme. Nitrogen is used in the reaction process to inhibit the oxidation reaction of tallow.

Oxidized tallow: The refined tallow processed by enzymatic hydrolysis (100 g) was put into a three-neck round-bottom flask and placed in an 120 °C oil bath. Stir at a speed of 200 r/min. When the temperature rises to 120 °C, and then air was injected at a speed of 1.2 L/min. Each sample were oxidized for 0 h (named T0), 1 h (named T1), 3 h (named T3) and 5 h (named T5), respectively.

### 3.3. FFAs Analysis

The FFAs was eluted, isolated and methylated according to the method in Song et al. with some modification [12]. Dissolve 0.8 g of tallow sample in 20 mL of n-hexane solution. An aminopropyl column (500 mg/6 mL Agilent Mega Bond Elut- NH2) was used to adsorb FFAs from tallow sample, then the glycerides were eluted with dichloromethane:isopropanol (2:1) solution, and finaly FFAs were eluted with 2% acetic acid-methyl tertbutyl ether solution.

The separated FFAs were concentrated into oily liquid by nitrogen. Then the methylation reaction was carried out by sulfuric acid method. After the reaction is completed and cooling to room temperature, hexane (2 mL), of water (3 mL) and 1 mL of internal standard (1 mL) were added and mixed thoroughly, where the internal standard solution is 3.6 mg/mL tridecanoic methyl-methanol solution. After standing for 12 h, take the supernatant for GC-MS analysis.

The FFAs were analyzed with Agilent 6890 GC equipped and a 5975 MS detector (Agilent Technologies, Santa Clara, CA, USA). A HP-INNOWax column (60 m × 0.25 mm i.d × 0.25 μm) was used to separate the FFAs. The flow rate of helium as carrier gas was 1 mL/min. The full-scan mode was adopted, with the electron ionization energy of MS was 70 eV and a mass range of 30–450 *m*/*z*, and the ion source temperature was 230 °C. The heating procedure is as follows: The oven temperature was maintained at 40 °C for 1 min, then increased to 120 °C at a rate of 10 °C/min and maintained for 3 min, increased to 190 °C at a rate of 10 °C/min, ramped at a rate of 2 °C/min to 230 °C, finally held for 15 min at 230 °C. The data in the NIST Mass Spectral Library 11 Vision was used to identified the FAME (fatty acid methyl ester) in tallow. Fatty acids were identified by the retention time of known standards. Tridecanoic methyl was selected as the internal standard and the relative quantitative correction factor and contents of FFAs were determined.

### 3.4. Volatile Compounds Analysis

#### 3.4.1. Isolation of the Volatiles by SAFE

40 g tallow sample (T0, T1, T3 and T5) was mixed with 400 μL internal standard (1,2-dichlorobenzene 100 ppm), respectively, and dissolved in 80 mL dichloromethane. The volatile compounds in the tallow samples were using the solvent-assisted flavor evaporation (SAFE) apparatus. Remove excess moisture with anhydrous sodium sulfate after the sample is dissolved, condense to 5.0 mL with rotary evaporator and finally to 1.0 mL by nitrogen.

#### 3.4.2. GC-MS Analysis

The volatile compounds in oxidized tallow were separated on two different polar columns: an HP-INNOWax analytical fused silica capillary column (60 m × 0.25 mm i.d × 0.25 μm) and a DB-5 analytical fused silica capillary column (60 m × 0.25 mm i.d × 0.25 μm). The model was used as FFA detection. The oven temperature was maintained at 40℃ for 3 min, then increased to 60 °C at a rate of 2 °C/min, increased to 120 °C at a rate of 4 °C/min, then increased to 140 °C at a rate of 2 °C/min, ramped at a rate of 5 °C/min to 230 °C, and finally held at 230 °C for 15 min.

#### 3.4.3. GC-O Analysis

Aroma Extract Dilution Analysis (AEDA) is used in the analysis of key aroma components. The concentrated samples obtained by SAFE were gradient diluted, such as 1:2, 1:4, 1:8, …, 1:1024. The experiment was stopped at unodourable dilution concentration. This experiment can determine the FD factor of aromatic compounds.

The GC-O analysis was performed on the GC-MS system equipped with an ODP-2 olfactory detection port (Gerstel, Mühlheim an der Ruhr, Germany). The aroma-active compounds with the same chromatographic conditions as GC-MS. The effluent of GC was split 1:1 between the flame ionization detector and the sniffing port, the double amount of injection was used (2 μL).

### 3.5. Identification and Quantification of the Volatile Compounds

The data in the NIST Mass Spectral Library 11 Vision was used to identified the volatile compounds in tallow. Retention indices were calculated using n-Alkanes (C7-C30) were analyzed on the polar (HP-INNOWax) and nonpolar (DB-5) columns under the same condition.

The standard curve was constructed for quantitative analysis of compounds with FD factor greater than 1. Refer to the method in, 50 mg of each of standard compound and 400 μL of internal standard solution to 40 g dichloromethane. The mix standard compounds solution was diluted with deionized water for five levels (1:5, 1:25, 1:50, 1:100 and 1:200) [38]. The standard curve was drawn by the plotting the response ratio of standard compounds and internal standard (Appendix A). All analyses were carried out in triplicate.

### 3.6. Determination of the Odor Activity Values (OAVs)

The compounds with OAVs ≥ 1 were identified as the key aroma-active component in oxidation tallow. OAV is the ratio of the concentration of aroma-active compounds to the threshold value in the sample. The threshold values were taken from the literature [24].

### 3.7. Sensory Analysis

The sensory profile was determined by sensory panel of 5 females and 5 males from Shanghai Institute of Technology, Shanghai, China. Sensory evaluation of tallow at different oxidation stages was carried out according to ISO 8589-2007 standard. The informed consents from the panelists have been obtained before the sensory experiments. Members of the sensory group received three sensory training sessions within two weeks, each with 2 h of training, and a total of 6 h.

Training session I. The members of the sensory panel evaluated the aroma characteristics of oxidized tallow and screened the sensory evaluation words.

Training session II. The purpose of this part was to get the panelists to reach a consensus on the aroma descriptors. Eight aroma descriptors were identified. Use physical references as indicators (Table 5).

Training session III. Training team members evaluated and discussed 8 selected sensory attributes in four samples of oxidized tallow in order to reach a consensus on the strength of perceptual attributes. Each aroma descriptors was rated on a scale of ten strengths (0–9).

Each tallow sample (5 g) was placed in a 15 mL sensory cup with a lid, which noted with a random number, in individual sensory booths at room temperature (25 °C) in a sensory room. The experience was carried out under dim light to mask the color difference between samples. And each group member then scored the samples, and there was a period of sensory recovery between the two samples. Each sample was repeated three times.

### 3.8. Statistical Analysis

The S-N-K test was applied to identify the significant differences (*p* ≤ 0.05) among individual samples for each aroma descriptor. SPSS 21 (SPSS Inc., Chicago, IL, USA) was used to evaluate sensory evaluation and GC-MS data by one-way analysis of variance (ANOVA). All statistics were analyzed by Excel (Microsoft, Redmond, WA, USA). The cluster heatmap was created using Origin 2018 software (OriginLab Corporation, Northampton, MA, USA). The Unscrambler version 9.7 (CAMO ASA, Oslo, Norway) was used for correlation analysis.

## 4. Conclusions

Oxidized tallow samples showed different sensory properties after different degrees of oxidation. Fatty (*p* ≤ 0.001), green (*p* ≤ 0.001), oxidized (*p* ≤ 0.001) and mutton (*p* ≤ 0.05) attributes demonstrated highly significant differences among the different stages of tallow oxidation. In total, 35 key aroma-active compounds whose OAVs were > 1. During tallow oxidation process, the concentration, FD and OAV of key aroma-active compounds all showed certain regularity. In general, 3 h of oxidation was considered to lead to the most prominent aroma. (*E*,*E*)-2,4-Heptadienal, (*E*,*E*)-2,4-decadienal, (*E*)-2-nonenal, octanal, hexanoic acid, hexanal and (*E*)-2-heptenal were found to be the key aroma-active compounds in oxidized tallow, whereas benzaldehyde was the key aroma-active compound in unoxidized tallow. PLSR analysis revealed the relationship of sensory attributes with FFAs and volatile flavor compounds in four stage of oxidized tallow, the contribution of aroma components and fatty acids to sensory attributes, and the effect of fatty acids as precursors on the flavor of oxidized tallow.

## Figures and Tables

**Figure 1 molecules-27-00352-f001:**
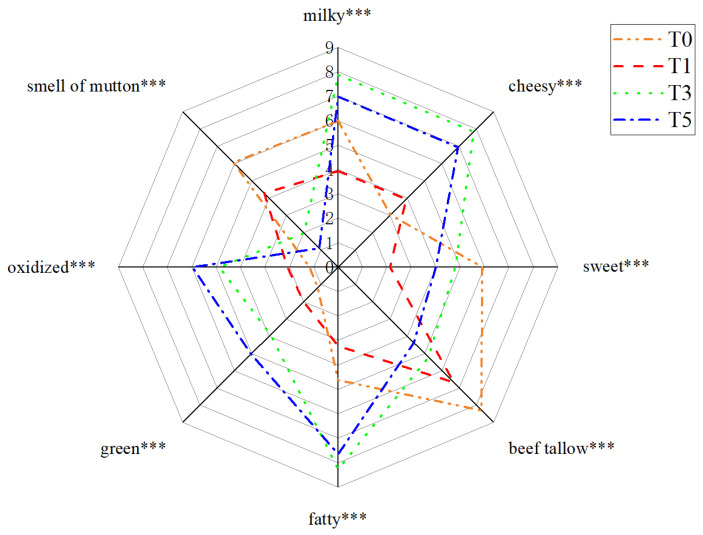
Sensory radar map of oxidized tallow for sensory evaluation. *** indicates significant at *p* ≤ 0.001.

**Figure 2 molecules-27-00352-f002:**
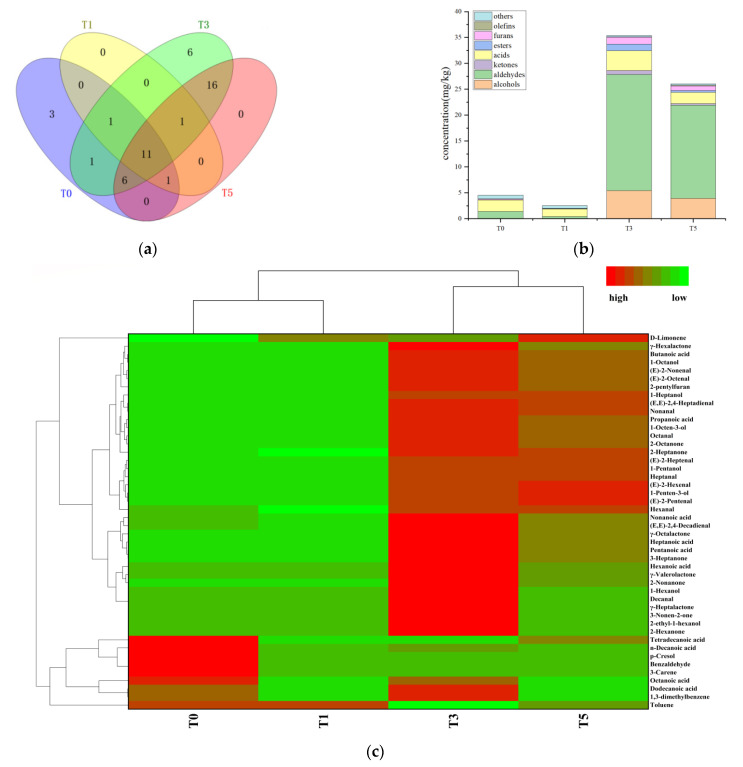
(**a**) volatile flavor compounds Venn diagram; (**b**) volatile flavor compounds concentration histogram; (**c**) volatile flavor compounds heatmap analysis.

**Figure 3 molecules-27-00352-f003:**
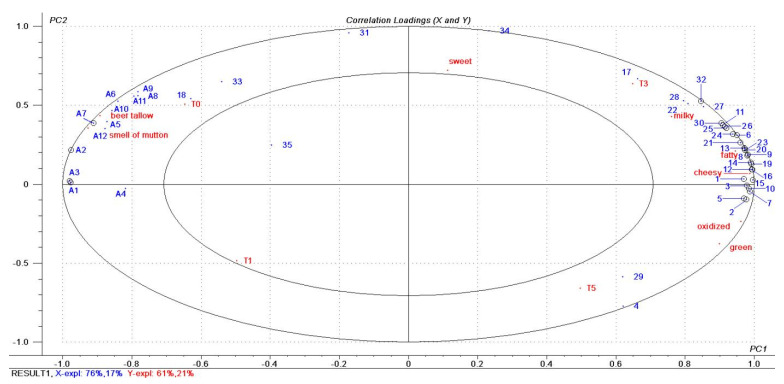
Correlation loading plot for FFAs and aroam−active compounds with sensory attributes in four stage tallow oxidation stages.

**Table 1 molecules-27-00352-t001:** Concentration of FFAs in the four oxidation stages of tallow.

No	FFA	Quality(mg/g) ^y^			
		T0 ^x^	T1	T3	T5
A1	C10:0 decanoic acid	0.12 ± 0.013 ^a^	0.10 ± 0.03 ^b^	0.05 ± 0.023 ^c^	0.07 ± 0.02 ^d^
A2	C11:0 undecaoic acid	0.025 ± 0.004 ^a^	0.02 ± 0.007 ^b^	0.007 ± 0.002 ^c^	0.005 ± 0.001 ^d^
A3	C12:0 lauric acid	0.09 ± 0.012 ^a^	0.07 ± 0.01 ^b^	0.04 ± 0.015 ^d^	0.05 ± 0.015 ^c^
A4	C14:0 myristic acid	1.21 ± 0.027 ^a^	0.99 ± 0.017 ^b^	0.78 ± 0.0015 ^c^	0.97 ± 0.018 ^b^
A5	C15:0 pentadecanoic acid	0.23 ± 0.017 ^a^	0.22 ± 0.02 ^a^	0.17 ± 0.012 ^b^	0.12 ± 0.02 ^c^
A6	C16:0 palmitic acid	10.29 ± 0.8421 ^a^	8.52 ± 0.87 ^b^	6.97 ± 0.89 ^c^	4.40 ± 0.12 ^d^
A7	C16:1 n-7 palmitoleic acid	0.62 ± 0.017 ^a^	0.03 ± 0.012 ^c^	0.04 ± 0.01 ^c^	0.21 ± 0.05 ^b^
A8	C17:0 heptadecanoic acid	1.06 ± 0.08 ^a^	0.87 ± 0.079 ^b^	0.67 ± 0.054 ^c^	0.16 ± 0.05 ^d^
A9	C18:0 stearic acid	8.68 ± 0.579 ^a^	6.80 ± 0.55 ^b^	5.50 ± 0.47 ^c^	1.85 ± 0.051 ^d^
A10	C18:1 n-9 oleic acid	6.76 ± 0.48 ^a^	5.55 ± 0.64 ^b^	3.57 ± 0.25 ^c^	1.46 ± 0.25 ^d^
A11	C18:2 n-6 linoleic acid	0.36 ± 0.025 ^a^	0.25 ± 0.005 ^b^	0.13 ± 0.07 ^c^	0.07 ± 0.013 ^d^
A12	C18:3 n-3 linolenic acid	0.03 ± 0.002 ^a^	0.03 ± 0.011 ^a^	0.01 ± 0.008 ^b^	-
A13	Trans palmitoleic acid	0.11 ± 0.003 ^c^	0.58 ± 0.012 ^a^	0.35 ± 0.09 ^b^	-
A14	Elaidic acid	0.14 ± 0.002 ^c^	0.23 ± 0.017 ^b^	0.35 ± 0.05 ^a^	0.0075 ± 0.002 ^d^
A15	Inoleic acid	-	0.05 ± 0.005 ^b^	0.09 ± 0.01 ^a^	-
A16	Other fatty acids	0.32 ± 0.09 ^b^	0.09 ± 0.02 ^c^	0.13 ± 0.04 ^c^	0.41 ± 0.1 ^a^
	Total	30.20 ± 2.3021	24.41 ± 2.305	18.90 ± 1.9955	9.76 ± 0.71

^x^ Four samples were denoted by the letter T followed by digit Arabic numbers, where Arabic numerals denote oxidation time (hours). Where T0 means refined tallow processed by enzymetic hydrolysis, T1, T3 and T5 were refined tallow treated by enzymatic hydrolysis–mild thermal oxidation. ^y^ Values bearing different superscripts (^a, b, c^ and ^d^) were significantly different (*p* < 0.05).

**Table 2 molecules-27-00352-t002:** Analysis of variance of the main effects and their interactions for each of the eight attributes in descriptive analysis.

Sensory Attribute	F-Values						Adjusted F-Value
	Samples(S) (df = 3)	Panelist(P) (df = 10)	Replication(R) (df = 1)	S × P (df = 30)	P × R (df = 10)	S × R (df = 3)	Sample ^a^ (S) (df = 30)
milky	796.10 ***	1.55	0.44	1.95 *	0.93	0.67	1159.201 ***
cheesy	1969.81 ***	2.19	0.25	1.52	0.97	1.35	2828.100 ***
sweet	228.73 ***	0.13	1.00	4.81 ***	0.03	1.00	651.280 ***
Beef tallow	187.14 ***	2.33	4.70	1.27	1.52	0.18	171.406 ***
fatty	137.85 ***	1.97	2.22	2.20 **	1.16	2.54 *	294.401 ***
green	589.50 ***	0.48	1.00	11.99 ***	0.16	0.21	1867.407 ***
oxidized	1170.95 ***	0.981	0.18	16.0 ***	1.11	1.29	4012.388 ***
Smell of mutton	1974.75 ***	2.10	0.26	4.65 ***	0.49	1.25	3955.130 ***

*, **, and *** indicate significant at *p* ≤ 0.05, *p* ≤ 0.01, and *p* ≤ 0.001, respectively. ^a^ Adjusted F-values of sample effect calculated using MSsample × panellistinstead of MSerroras described in the text.

**Table 3 molecules-27-00352-t003:** FD factors of volatile flavor compounds in four samples.

No	Compounds	FD Factors			
		T0	T1	T3	T5
1	Hexanal	8	32	128	128
2	1-Penten-3-ol	0	0	2	2
3	Heptanal	8	4	8	4
4	D-Limonene	0	16	1	1
5	(*E*)-2-Hexenal	2	2	8	8
6	2-pentylfuran	2	2	8	4
7	1-Pentanol	0	0	32	4
8	2-Octanone	0	0	32	1
9	Octanal	2	64	256	256
10	(*E*)-2-Heptenal	8	64	128	256
11	2-Nonanone	0	0	32	4
12	Nonanal	8	4	32	16
13	(*E*)-2-Octenal	0	0	8	32
14	1-Octen-3-ol	0	0	64	64
15	1-Heptanol	0	0	4	256
16	(*E*,*E*)-2,4-Heptadienal	1	256	1024	512
17	Decanal	0	0	2	0
18	Benzaldehyde	64	0	0	0
19	Propanoic acid	0	0	64	64
20	(*E*)-2-Nonenal	16	512	512	512
21	1-Octanol	0	0	64	1
22	γ-Valerolactone	0	0	0	4
23	Butanoic acid	0	0	64	16
24	γ-Hexalactone	0	0	2	4
25	Pentanoic acid	0	0	8	4
26	γ-Heptalactone	0	0	2	0
27	(*E*,*E*)-2,4-Decadienal	4	32	512	512
28	Hexanoic acid	4	32	512	256
29	γ-Octalactone	0	0	0	4
30	Heptanoic acid	0	4	8	4
31	Octanoic acid	8	16	16	16
32	Nonanoic acid	0	0	1	0
33	n-Decanoic acid	16	0	1	1
34	Dodecanoic acid	2	32	1	4
35	Tetradecanoic acid	16	14	64	128

**Table 4 molecules-27-00352-t004:** Concentrations and OAVs of aroma-active compounds in four samples.

No	Compounds	RI		Odor Discreption	Concentration (mg/kg) ^c^				Threshold (mg/kg) ^d^	OAV				Identification ^e^
		HP-INNOWax ^a^	DB-5 ^b^		T0	T1	T3	T5		T0	T1	T3	T5	
1	Hexanal	1085	-	green, fatty	30.9 ± 1.13 ^c^	9.0 ± 0.87 ^d^	111.0 ± 1.5921 ^b^	116.2 ± 0.9852 ^a^	0.005 ^a^	6174	1791	22,196	23,232	MS,RI,O
2	1-Penten-3-ol	1153	-	green, pungent	-	-	22.8 ± 0.89 ^b^	26.2 ± 0.245 ^a^	0.3581 ^a^	-	-	64	73	MS,RI,O
3	Heptanal	1182	902	fatty, green, herbal	98.6 ± 2.6519 ^c^	49.0 ± 1.2 ^d^	585.0 ± 10.2378 ^b^	613.9 ± 8.5513 ^a^	0.0028 ^a^	35,200	17,493	208,918	219,257	MS,RI,O
4	D-Limonene	1192	1032	citrus, sweet, peel	-	5.7 ± 0.04 ^b^	4.6 ± 0.0311 ^c^	8.9 ± 0.0654 ^a^	0.034 ^a^	-	167	135	262	MS,RI,O
5	(*E*)-2-Hexenal	1211	855	green, fruity	9.0 ± 0.038 ^c^	4.5 ± 0.08 ^d^	90.7 ± 1.2306 ^b^	105.9 ± 1.003 ^a^	0.0028 ^a^	102	51	1024	1196	MS,RI,O
6	2-pentylfuran	1224	990	fruity, green	15.8 ± 0.69 ^c^	4.4 ± 0.08 ^d^	95.3 ± 1.5546 ^a^	61.4 ± 0.7845 ^b^	0.0058 ^a^	2725	759	16,423	10,590	MS,RI,O
7	1-Pentanol	1236	-	fermented, pungent	-	-	145.4 ± 4.23 ^b^	152.7 ± 0.8894 ^a^	0.1502 ^a^	-	-	968	1016	MS,RI,O
8	2-Octanone	1273	-	earthy, musty, cheesy	2.9 ± 0.035 ^c^	-	27.3 ± 0.8974 ^a^	21.2 ± 0.1457 ^b^	0.0502 ^a^	57	-	543	422	MS,RI,O
9	Octanal	1278	1002	green	52.3 ± 0.68 ^c^	12.2 ± 0.17 ^d^	549.6 ± 8.841 ^a^	410.7 ± 4.4437 ^b^	0.000587 ^a^	89,064	20,853	936,344	699,633	MS,RI,O
10	(*E*)-2-Heptenal	1310	958	green, fatty, fruity	17.5 ± 0.3458 ^c^	-	401.3 ± 7.563 ^b^	420.8 ± 5.1126 ^a^	0.051 ^a^	344	-	7868	8252	MS,RI,O
11	2-Nonanone	1371	1990	Fruity, sweet, waxy	-	-	14.9 ± 0.2113 ^a^	6.4 ± 0.031 ^b^	0.041 ^a^	-	-	364	157	MS,RI,O
12	Nonanal	1377	1104	fatty, citrus, green	7.6 ± 0.623 ^c^	-	214.5 ± 2.31 ^a^	181.7 ± 1.2897 ^b^	0.0011 ^a^	6887	-	195,013	165,225	MS,RI,O
13	(*E*)-2-Octenal	1412	1059	fatty, green, herbal	-	-	321.9 ± 4.332 ^a^	202.5 ± 1.9613 ^b^	0.003 ^a^	-	-	107,304	67,503	MS,RI,O
14	1-Octen-3-ol	1423	-	earthy, mushroom, green	-	-	89.8 ± 1.5843 ^a^	67.4 ± 0.5633 ^b^	0.0015 ^a^	-	-	59,856	44,928	MS,RI,O
15	1-Heptanol	1427	971	green, musty, pungent	-	-	216.7 ± 5.6617 ^a^	200.2 ± 1.4785 ^b^	0.0054 ^a^	-	-	40,124	37,073	MS,RI,O
16	(*E*,*E*)-2,4-Heptadienal	1447	1011	fatty, green, oily	-	-	111.0 ± 1.589 ^a^	84.3 ± 0.8521 ^b^	0.057 ^b^	-	-	1803	1479	MS,RI,O
17	Decanal	1483	1205	sweet, aldehydic	-	-	53.8 ± 1.22 ^a^	-	0.003 ^a^	-	-	17,936	-	MS,RI,O
18	Benzaldehyde	1507	963	fruity, almond, sweet	0.7 ± 0.081 ^a^	-	-	-	0.75089 ^a^	1	-	-	-	MS,RI,O
19	Propanoic acid	1514	-	acidic, pungent, dairy	-	-	35.1 ± 0.8524 ^a^	26.9 ± 0.9971 ^b^	2.19 ^a^	-	-	16	12	MS,RI,O
20	(*E*)-2-Nonenal	1526	1158	fatty, green, cucumber	3.9 ± 0.1124 ^d^	10.8 ± 0.51 ^c^	334.5 ± 3.872 ^a^	212.4 ± 2.3001 ^b^	0.00019 ^a^	20,749	56,751	1,760,642	118,026	MS,RI,O
21	1-Octanol	1533	1070	waxy, green, citrus	-	-	243.1 ± 7.1102 ^a^	140.5 ± 1.7736 ^b^	0.1258 ^a^	-	-	1932	1117	MS,RI,O
22	γ-Valerolactone	1600	951	herbal, sweet, woody	-	-	45.6 ± 0.9857 ^a^	10.8 ± 0.054 ^b^	-	-	-	-	-	MS,RI,O
23	Butanoic acid	1606	791	cheesy, dairy, buttery	-	-	19.4 ± 0.052 ^a^	12.1 ± 0.03 ^b^	2.4 ^a^	-	-	8	5	MS,RI,O
24	γ-Hexalactone	1693	1051	tonka, sweet, creamy	-	-	53.7 ± 0.8864 ^a^	27.1 ± 0.4613 ^b^	0.26 ^a^	-	-	207	104	MS,RI,O
25	Pentanoic acid	1716	879	cheesy, acidic,	-	-	35.1 ± 0.7985 ^a^	33.4 ± 0.7763 ^b^	11 ^a^	-	-	7	3	MS,RI,O
26	γ-Heptalactone	1795	1148	coconut, sweet, coumarinic	-	-	28.0 ± 0.4562 ^a^	12.4 ± 0.1523 ^b^	0.4 ^a^	-	-	70	31	MS,RI,O
27	(*E*,*E*)-2,4-Decadienal	1802	1320	fatty	11.4 ± 0.8887 ^c^	-	130.2 ± 3.2543 ^a^	47.0 ± 0.7134 ^b^	0.000027 ^a^	421,063	-	4,821,417	1,740,885	MS,RI,O
28	Hexanoic acid	1824	975	fatty, sour, sweet, cheesy	40.5 ± 2.17 ^c^	38.8 ± 0.69 ^c^	153.2 ± 2.3214 ^a^	60.6 ± 0.5684 ^b^	0.89 ^a^	46	44	152	68	MS,RI,O
29	γ-Octalactone	1910	1253	coconut, sweet	-	-	17.6 ± 0.3254 ^a^	4.5 ± 0.081 ^b^	0.0156 ^a^	-	-	1131	289	MS,RI,O
30	Heptanoic acid	1930	1067	cheesy, sweet	8.8 ± 0.62 ^c^	8.6 ± 0.04 ^c^	20.8 ± 0.6231 ^a^	13.9 ± 0.0125 ^b^	0.64 ^a^	14	13	33	22	MS,RI,O
31	Octanoic acid	2033	1160	fatty, cheesy	68.8 ± 3.01 ^a^	49.8 ± 1.11 ^c^	64.0 ± 0.8787 ^b^	49.0 ± 0.21 ^c^	3 ^a^	23	17	21	16	MS,RI,O
32	Nonanoic acid	2136	1257	waxy, cheesy	8.9 ± 0.58 ^c^	-	27.2 ± 0.6613 ^a^	15.0 ± 0.02 ^b^	4.6 ^a^	2	-	6	3	MS,RI,O
33	n-Decanoic acid	2243	1357	fatty, citrus	51.4 ± 1.64 ^a^	34.5 ± 0.95 ^c^	37.3 ± 0.7461 ^b^	34.9 ± 0.7741 ^c^	10 ^a^	5	3	4	3	MS,RI,O
34	Dodecanoic acid	2461	1555	fatty, coconut	17.5 ± 0.77 ^b^	15.0 ± 0.24 ^c^	18.9 ± 0.058 ^a^	15.0 ± 0.085 ^c^	0.1 ^b^	175	150	189	150	MS,RI,O
35	Tetradecanoic acid	2679	-	waxy, fatty, citurs	30.7 ± 1.09 ^a^	23.2 ± 0.77 ^c^	23.7 ± 0.7755 ^c^	26.8 ± 1.008 ^b^	10 ^a^	96	72	74	84	MS,RI,O

^a^ RI of compounds on an HP-INNOWax column. ^b^ RI of compounds on a DB-5 column. ^c^ Mean ± standard deviation for four tallow samples. Means in the same row with different superscripts (^a, b, c^ and ^d^) are significantly different (*p* ≤ 0.05). ^d^ Means odor thresholds were unavailable. ^a^ Means odor threshold in water according to Van Germert (2011). ^b^ Means odor threshold in air according to Van Germert (2011). ^e^ Identification method: MS means identification by comparison with the NIST 11 mass spectra database; RI means confirmed by comparison of RI to reference standards; O means confirmed by aroma descriptor.

**Table 5 molecules-27-00352-t005:** The corresponding reference of oxidized tallow sensory descriptor.

Aroma Description	Reference Sample
Milky	Fresh milk
Cheesy	cheesy
Sweet	sweets
Beef tallow	tallow
Fatty	Animal fat
Green	Freshly cut grass
Oxidized	Fresh milk (ultraviolet light 12 h)
Smell of mutton	Beef

## Data Availability

Data is contained with in the article.

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
