# Peer review of "Evolution Analysis of Free Fatty Acids and Aroma-Active Compounds during Tallow Oxidation"

_molecules, 2022, doi:10.3390/molecules27020352_

Round 1

Reviewer 1 Report

1)The manuscript could still provide some information on the odor-active compounds in the commodity.

2)The organization of the introduction needs to be improved, it will be helpful to get professional proof-editing to improve the readability.

3)How to select the threshold in OAV calculation.

Author Response

Dear Reviewer,

We are truly grateful to you for giving us the chance to revise our manuscript. Based on comments and suggestions from you, we have made careful modifications on the revised manuscript.  The point-by-point responses to the comments are appended below.

1)The manuscript could still provide some information on the odor-active compounds in the commodity.

Response:

We have supplemented some information on the odor-active compounds in section “2.5 Evolution analysis of aroma-active compound in oxidized tallow”.

2)The organization of the introduction needs to be improved, it will be helpful to get professional proof-editing to improve the readability.

Response:

Thank you for your favorable comments on our research. We have asked an English native speaker to edit the manuscript to improve the readability.

3)How to select the threshold in OAV calculation.

Response:

The thresholds for these compounds are referenced Ref. 25, which is an odor threshold book written by L.J. van Gemert. This book summarizes the odor thresholds of compounds measured by most researchers until 2011. These thresholds include absolute thresholds and difference threshold. The detection and the recognition thresholds are absolute thresholds. The first being the minimum concentration which can be detected without any requirements to identify or recognize the stimulus, while the second one is the minimum concentration at which a stimulus can be identified or recognized. In our study, the detection thresholds was used as the standard for OAV calculation. In addition, the threshold value of compounds in food is usually based on the threshold value in water, so the detection threshold value in water were chosen in this study.

The manuscript has been resubmitted to your journal. We are looking forward to your positive response.

Best regards,

Min Sun

Reviewer 2 Report

I highly recommend the article for publication in the reputed journal MDPI Molecules 

I have gone through the manuscript and am pleased with the information and English language style provided.   

Authors have carried out solid phase extraction (SPE) coupled with gas chromatography mass and spectrometry for assessment of free  fatty acid in the oxidation process of tallow of different sources. Besides, solvent-assisted flavor evaporation together with GC-MS and with olfactometry was also examined for estimation of the key aroma-active compounds generated in the oxidized tallow. The statistical analysis was carried out with the tool “Partial least-squares regression” for the obtained results clubbed with sensory evaluation scores. Authors have also revealed the information pertinent to the formation mechanism of characteristic flavor of oxidized tallow, explaining the quantitative effect between characteristic meat flavor precursor compounds. The authors have also revealed the synthesis pathways of the different flavor precursors engaged in the oxidation and deciphered underlying mechanisms involved in the lipid oxidation. The authors have concluded the study with revealing changes in free fatty acid along with identification of volatile compounds in four oxidation processes of tallow. Simultaneously, authors have summarized the important  aroma active compounds involved in fatty acid oxidation and their metabolic evolution in the oxidation process.  

Hence, I strongly recommend the manuscript for the publication.   

Author Response

Dear Reviewer,

Thank you for your favorable comments on our research. We have asked an English native speaker to edit the manuscript to make sentences easier to understand.

The manuscript has been resubmitted to your journal. We are looking forward to your positive response.

Best regards,

Min Sun
